# Integrated Microbiome–Metabolome Analysis and Functional Strain Validation Reveal Key Biochemical Transformations During Pu-erh Tea Pile Fermentation

**DOI:** 10.3390/microorganisms13081857

**Published:** 2025-08-08

**Authors:** Mengkai Hu, Huimin Zhang, Leisa Han, Wenfang Zhang, Xinhui Xing, Yi Wang, Shujian Ou, Yan Liu, Xiangfei Li, Zhenglian Xue

**Affiliations:** 1Engineering Laboratory for Industrial Microbiology Molecular Beeding of Anhui Province, College of Biologic & Food Engineering, Anhui Polytechnic University, 8 Middle Beijing Road, Wuhu 241000, China; mengkaihu@ahpu.edu.cn (M.H.); zhm_catharine@126.com (H.Z.); hanleisa123@163.com (L.H.); 17856691286@163.com (W.Z.); liuyan@ahpu.edu.cn (Y.L.); 2Department of Chemical Engineering, Tsinghua University, Beijing 100084, China; xhxing@tsinghua.edu.cn (X.X.); wangyi671@tsinghua.edu.cn (Y.W.); 3Yunnan Shujian Tea Co., Ltd., Xishuangbanna 650501, China; 15295582534@163.com

**Keywords:** Pu-erh tea, pile fermentation, microbial communities, non-targeted metabolomics analysis, macromolecule degradation

## Abstract

Fermentation plays a pivotal role in shaping the flavor and overall quality of Pu-erh tea, a microbially fermented dark tea. Here, we monitored physicochemical properties, chemical constituents, and microbial succession at 15 fermentation time points. Amplicon sequencing identified *Staphylococcus*, *Bacillus*, *Kocuria*, *Aspergillus*, *Blastobotrys*, *Thermomyces*, and *Rasamsonia* as dominant genera, with prokaryotic communities showing greater richness and diversity than eukaryotic ones. Beta diversity and clustering analyses revealed stable microbial structures during late fermentation stages. Non-targeted metabolomics detected 347 metabolites, including 56 significantly differential compounds enriched in caffeine metabolism and unsaturated fatty acid biosynthesis. Fermentation phases exhibited distinct metabolic patterns, with volatile aroma compounds (2-acetyl-1-pyrroline, 2,5-dimethylpyrazine) and health-beneficial fatty acids (linoleic acid, arachidonic acid) accumulating in later stages. OPLS-DA and KEGG PATHWAY analyses confirmed significant shifts in metabolite profiles relevant to flavor and biofunctionality. RDA revealed strong correlations between microbial taxa, environmental parameters, and representative metabolites. To functionally verify microbial contributions, 17 bacterial and 10 fungal strains were isolated. Six representative strains, mainly *Bacillus* and *Aspergillus*, exhibited high enzymatic activity on macromolecules, confirming their roles in polysaccharide and protein degradation. This integrative multi-omics investigation provides mechanistic insights into Pu-erh tea fermentation and offers a scientific basis for microbial community optimization in tea processing.

## 1. Introduction

Tea has historically held a significant position in China, fostering the development of a rich tea culture [1]. Based on their fermentation degrees, Chinese tea varieties can be categorized into six types, namely green tea, white tea, yellow tea, oolong tea, black tea, and dark tea, with darker soups indicating higher fermentation levels. Dark tea, as a post-fermented variety, undergoes five key steps, namely withering, kneading, fixing, drying, and piling, with the latter stage imparting a characteristic black color to the leaves [2]. Notable dark tea varieties include Anhui old dark tea, Yunnan Pu-erh tea, Hunan black tea, Hubei green brick tea, Guangxi Liupu tea, Shaanxi Fu tea, and Sichuan Tibet tea, among which Yunnan Pu-erh tea enjoys global acclaim [3]. Produced from Yunnan *Camellia* sinensis var. assamica, Pu-erh tea is distinguished by its unique woodiness, mellow flavor, low astringency, and vibrant red soup color [4]. Furthermore, its health benefits—including anti-obesity, anti-hyperglycemia, anti-hyperlipidemia, antioxidant, and anti-cancer properties—are well documented [5,6,7,8]. Importantly, the rich nutritional profile and unique taste of Pu-erh tea are closely linked to the pile fermentation process, which significantly influences the final quality. Therefore, exploring the intricacies of this fermentation method is essential in ensuring consistent quality in traditional production practices.

The pile fermentation of Pu-erh tea involves a complex interplay among microorganisms, serving as the key to unlocking the fermentation dynamics of cooked tea [9]. Deconstructing the microbial ecosystem involved in Pu-erh tea fermentation poses significant challenges. First, the microbial sources are diverse and largely uncontrollable, originating from the tea leaves, water, and surrounding environment. The alterations in the microbial community structure during fermentation critically influence the flavor composition and sensory attributes. Second, pile fermentation involves the restructuring of tea leaves, driven by both prokaryotic and eukaryotic microorganisms. However, the specific species and metabolic functions of the core functional microbes in Pu-erh tea remain poorly understood, with the impact of the tea content on microbial metabolism often overlooked. Moreover, various factors, such as the temperature, moisture content, water activity, and pH, significantly affect the fermentation process, subsequently influencing microbial growth and metabolism, which in turn impact the quality of Pu-erh tea. While some research has advanced our understanding of Pu-erh tea, much of it focuses on flavor profiles, including the effects of different fermentation materials and degrees on tea quality. Unfortunately, studies detailing the dynamic processes of Pu-erh tea fermentation are limited [10]. Previous studies on Pu-erh tea fermentation have focused on microbial community profiling and flavor development [11,12,13]. However, these studies have often employed single-omics or descriptive methods. For instance, while Li et al. and Zhao et al. examined the microbial composition and chemical changes, they did not explore the functional contributions of the microbiota to specific metabolic pathways or the interactions between the microbiota and non-volatile metabolites during fermentation [11,13]. Thus, it is essential to elucidate the changes in the microbial community structure, physical and chemical parameters, and flavor substance trends during fermentation, thereby establishing a resource base for a natural strain pool and enhancing the functional study of dominant microbes to improve the quality stability of Pu-erh tea fermentation.

Advancements in biotechnology and bioinformatics, particularly high-throughput sequencing (HTS) technology and data analysis methods, have significantly facilitated research in fermented foods. The role of metabolomics in this field is becoming increasingly prominent, with applications evident in products such as koji, yogurt, and kimchi [14,15,16]. These technologies enable researchers to bypass traditional culturing methods, revealing the microbial compositions of samples at various stages of fermentation through sequencing and allowing for the exploration of key factors influencing microbial communities via association analysis. Consequently, the structures and functional genes of microbial communities can be systematically investigated. Additionally, comprehensive analyses of compositional changes, phylogenetic evolution, and the potential functional genes of microorganisms during fermentation shed light on the complexities of the microbial communities in fermented foods, thereby aiding the establishment of stable and standardized fermentation processes.

In this study, we employed throughput sequencing microbiome techniques and LC–Orbitrap/MS-based untargeted metabolomics to investigate the microbial community dynamics, metabolite variations, and physicochemical changes during the 30-day pile fermentation of Pu-erh tea. Multi-omics integration was applied to analyze the correlations between microbial taxa, environmental parameters, and key metabolites involved in flavor development. Furthermore, representative prokaryotic and fungal strains were isolated and functionally assessed for their enzymatic activity in degrading starch, pectin, cellulose, and protein. This combined approach aimed to elucidate the microbial mechanisms underlying biochemical transformations in Pu-erh tea, while laying a foundation for microbial intervention strategies to improve fermentation control and product quality.

## 2. Materials and Methods

### 2.1. Tea Samples and Reagents

Samples of Yunnan Little Brown Pu-erh tea were provided by Yunnan Shujian Tea Co., Ltd. (Xishuangbanna, China). The primers used for library construction prior to high-throughput sequencing were synthesized at Nanjing GenScript Biotechnology Co., Ltd. (Nanjing, China). The primer sequences are presented in Appendix A. HPLC-grade acetonitrile and formic acid were purchased from Thermo Fisher (Waltham, MA, USA). Other reagents, such as ninhydrin, rutin, sulfuric acid, and methanol, were purchased from the Sinopharm Group (Shanghai, China), unless otherwise noted.

### 2.2. Tea Sampling

On 19 August 2022 (day 0 of fermentation), fresh leaves of Yunnan Little Brown Pu-erh tea were selected for pile fermentation at Yunnan Shujian Tea Co., Ltd. (Xishuangbanna, China), according to the process shown in Appendix A. Fermentation was completed on 18 September 2022 (the 30th day), and then the leaves were used to produce fermented Pu-erh tea. Fifteen key nodes in the fermentation process were selected for sampling: pre-fermentation (OD), the first fermentation cycle (1D, 2D, 3D, 5D, 6D), the second fermentation cycle (10D, 12D, 14D), the third fermentation cycle (18D, 20D. 22D), the fourth fermentation cycle (26D), the fifth fermentation cycle (28D), and the fermentation end-point (30D). Six groups of tea samples were collected each time. Then, 200 g of tea leaves (Appendix A) was selected from each of 5 points, A, B, C, D, and E, in the middle layer of the tea stack; they were then mixed in sterile bags and stored at −20 °C until testing. Sample information can be found in Appendix A. This sampling strategy was based on the assumption that the middle layer of the fermentation pile would provide a representative microbial and physicochemical profile. Previous studies have indicated that this region exhibits relatively stable temperature, moisture, and oxygen conditions compared to the top and bottom layers, thereby minimizing sampling bias.

### 2.3. Metagenome DNA Extraction, Library Construction, and High-Throughput Sequencing of Amplicons of Pu-erh Tea Samples

DNA samples were extracted using a DNA kit (M5635-02) (Omega Bio-Tek, Norcross, GA, USA) and used as the templates for library construction. Then, the V3-V4 region of prokaryotic 16S rRNA was amplified using the 338F/806R primer pair, while the ITS1F/ITS2R primer pair was used to amplify the ITS1 region in the eukaryotic case. High-throughput amplicon sequencing was carried out on the NovaSeq6000 platform at Shanghai Personalbio Biotechnology Co., Ltd. (Shanghai, China). The primer sequences are provided in Appendix A.

### 2.4. Raw Data Processing and Analysis of High-Throughput Sequencing

The raw amplicon sequencing data were processed using QIIME2 for quality filtering, denoising, splicing, and chimera removal, resulting in high-quality ITS and 16S rRNA sequences. Operational taxonomic units (OTUs) and prokaryotic amplicon sequence variants (ASVs) were generated by clustering the high-quality sequences. OTUs, similarly to ASVs, are commonly used to represent microbial richness and diversity [17,18]. Eukaryotic OTUs and prokaryotic ASVs were assigned taxonomic identities via comparison with the UNITE database (Release 8.0) and the SILVA database (Release 132), respectively, resulting in abundance tables for downstream analysis. Alpha diversity indices were calculated using QIIME2, and the statistical significance of the differences in α-diversity among samples was assessed by ANOVA. Principal component analysis (PCA) was performed via the R language to analyze the differences in the flora structure among the samples. RDA correlation analysis was performed via the Canoco 5 software.

### 2.5. Non-Targeted Metabolomics Profiling

#### 2.5.1. Pretreatment of Tea Samples

Tea samples were first freeze-dried, homogenized, and passed through a 100-mesh sieve. A total of 0.4 g of the resulting powder was extracted with 8 mL of 70% methanol by ultrasonic treatment for 30 min, followed by shaking at room temperature for 4 h. After extraction, 1.5 mL of the supernatant was transferred into a 2 mL centrifuge tube and centrifuged at 12,000 rpm for 10 min. For quality control (QC), 30 μL of the supernatant from each sample was pooled in an EP tube.

An additional 100 μL of the supernatant was also removed and mixed with 3.9 mL of 70% methanol for dilution, which was repeated. Finally, 1 mL of the above diluent was centrifuged at 12,000 rpm for 10 min at 4 °C. The resulting supernatant was filtered through a 0.22 μm membrane, and the filtrate was transferred to a liquid vial for analysis by liquid chromatography–orbital ion trap mass spectrometry (LC–Orbitrap/MS). Each sample was prepared in 6 parallels.

#### 2.5.2. Liquid Chromatography–High-Resolution Mass Spectrometry (LC-HRMS) Conditions

Non-targeted metabolomics was entrusted to Shanghai BIOTREE Biotech Co., Ltd. (Shanghai, China). The Pu-erh tea samples with various fermentation times were analyzed using LC–Orbitrap/MS with the ΜLtimate 3000 high-performance liquid chromatography device and the Thermo Fisher Q-Exactive Focus mass spectrometer (Waltham, MA, USA). For LC analysis, a Thermo Hypersilgold (Waltham, MA, USA) (100 mm × 2.1 mm, 1.9 μm) device was employed. Eluent A was 0.1% formic acid, while eluent B was a 0.1% formic acid–acetonitrile mixture. The flow rate for gradient elution was set at 0.4 mL/min, and the column temperature was kept at 40 °C. The gradient was as follows: 0~10 min 98% A, 10~11 min 50% A, 11~13 min 5% A, and 13~15 min 98% A. The samples were detected at a 210 nm wavelength.

For MS analysis, the data acquisition mode was Full Mass Spectrometry–Direct Detection Mass Spectrometry 2 (Full MS-ddMS2). The ion source was HESI+/−. The spray voltage was +3400/−3400 V, and the temperature of the ion transfer tube was 350 °C, with 45 arb sheath gas and 15 arb auxiliary air. The mass-to-charge ratio ranged from 100 to 1500 *m*/*z*. The Full MS resolution was 7000; the ddMS2 resolution was 17,500. The fragmentation mode was high-energy collision dissociation (HCD), with normalized collision energy (NCE) of 30, 40, and 60.

#### 2.5.3. Data Processing and Analysis of Non-Targeted Metabolomics

Raw HPLC-MS data were acquired using the SCIEX Analyst Workstation software (1.6.3). Mass spectrometry files were converted to mzXML format using ProteoWizard, and data matrices were generated using the MAPS software (1.0). To preprocess the data, outliers were filtered, and missing values were imputed using half the minimum value. The total ion current (TIC) of each sample was standardized and normalized. Finally, the material peaks were annotated via the Human Metabolome Database (HMDB) and Kyoto Encyclopedia of Genes and Genomes (KEGG).

Principal coordinate analysis (PCoA) and orthogonal partial least squares discriminant analysis (OPLS-DA) were conducted via the SIMCA software (V16.0.2). A permutation test was performed within OPLS-DA to verify the validity of the model. The *p*-value obtained by Student’s *t*-test and the variable importance in projection (VIP) (*p* < 0.05 and VIP > 1) of the first principal component of the OPLS-DA were used as the criteria for the screening of differential metabolites between groups. The metabolic pathways associated with differential metabolism were annotated via the KEGG PATHWAY database.

Relative abundance charts and bar plots were generated using Origin 8.5. SPSS 27.0 and analysis of variance (ANOVA) were used to analyze the significance of the differences between samples. Redundancy analysis (RDA) was conducted using Canoco 5.0 to explore relationships among microbial communities, physicochemical parameters, and flavor compounds.

### 2.6. Determination of Biochemical Parameters of Pu-erh Tea

The temperature was measured using an electronic thermometer inserted diagonally into the middle layer of the tea pile, and three groups were measured each time.

For pH determination, tea leaves were first dried, crushed, and sieved. Then, 1 g of the resulting powder was mixed with 10 mL of deionized water and allowed to stand for 20 min before being measured using a pH meter.

The determination of moisture content was carried out in a drying dish, drying 5 g (accurate to 0.001 g) of tea in an oven at 103 °C. To ensure consistency, weighing was repeated until the difference between two consecutive measurements was less than 0.005 g.

The water activity (Aw) of Pu-erh tea was detected in the fermentation chamber using a water activity meter. Prior to measurement, sample cups were wiped clean with Kimwipes and placed horizontally. Pu-erh tea samples were cut and evenly spread in the cup, ensuring that the tea only covered the bottom, with a thickness of less than half the depth of the cup.

### 2.7. Determination of Main Chemical Components of Pu-erh Tea

The concentration of soluble polysaccharides was determined using the anthrone–sulfuric acid colorimetric method. In this assay, sugars are dehydrated by concentrated sulfuric acid to form furfural and its derivatives, which subsequently react with anthrone to produce a blue-colored complex, with maximum absorbance at 620 nm. Briefly, 0.2 g of tea powder was dissolved in 20 mL of deionized water and incubated at 100 °C for 1 h. The supernatant was then collected and filtered. This extraction step was repeated three times, and all filtrates were combined. Subsequently, 240 mL of absolute ethanol was added to the pooled filtrate, and the mixture was stored at 4 °C for 24 h to allow precipitation. Then, it was centrifuged at 8000 rpm for 10 min and the supernatant was discarded. The precipitate was re-dissolved and diluted 200 times with deionized water. Then, 1 mL of the solution was pipetted into another centrifuge tube, and 4 mL of anthone test solution (200 mg of anthone, 100 mL of sulfuric acid solution) was added. The reaction mixture was incubated at 100 °C for 6 min and then cooled in an ice water bath for 15 min; finally, the value of each sample was recorded at 620 nm.

Flavonoid content was determined using the aluminum chloride colorimetric method, with spectrophotometric measurements at 415 nm. This method is based on the formation of a coordination complex between the hydroxyl groups of flavonoids and aluminum chloride [19,20]. Briefly, 1.0 g of tea powder was extracted with 50% ethanol at a solid-to-liquid ratio of 1:70 in a water bath at 80 °C for 5 h. After filtration, 5 mL of the extract was transferred into a 25 mL colorimetric tube, followed by the addition of 8 mL of 1.5% AlCl_3_ solution (pH 8.0) and 4 mL of acetate–sodium acetate buffer (pH 5.5). The mixture was then diluted to 25 mL with 50% ethanol. After standing for 30 min at room temperature, the absorbance was measured at 415 nm.

The theabrownin content was determined according to the colorimetric extraction method. First, 3 g of tea leaves was boiled in 125 mL of distilled water for 10 min; 25 mL of the filtrate was then added to 25 mL of n-butanol and shaken for 3 min. Afterwards, 2 mL of the aqueous phase was collected and diluted to 25 mL with 2 mL of saturated oxalic acid solution, 6 mL of water, and 15 mL of 95% ethanol, and, finally, the wavelength was measured at 380 nm.

Cellulose and lignin concentrations were determined using the concentrated sulfuric acid method, as previously described [21]. The procedure for the measurement of the cellulose content in tea is detailed as an example. First, 0.10 g of tea powder was added to 5 mL of acetic acid/nitric acid (1:1, *v*/*v*) and heated in a boiling water bath for 30 min. After cooling, the supernatant was removed by centrifugation at 8000 rpm for 5 min. After washing twice with distilled water, the precipitate was dried in an oven at 80 °C. Next, 10 mL of the mixture (10% sulfuric acid and 0.1 mol/L potassium dichromate) was added to the precipitate and it was cooled. Then, 5 mL of 20% KI was added, as well as 5 mL of 0.5% starch solution, before titration. Finally, the solution was titrated with 0.2 mol/L sodium thiosulfate until it was blue and did not change color within half a minute, and another blank was performed without adding a sample.

### 2.8. Determination of Polyphenols, Protein, and Free Amino Acid Concentration

The content of tea polyphenols, protein, and free amino acids was determined by the Folin–Ciocalteu, Kjeldahl, and ninhydrin colorimetry methods, respectively, as described in the literature [22,23].

### 2.9. Evaluation of Hydrolytic Capabilities of Representative Strains on Solid Media

To evaluate the enzymatic capabilities of representative strains in degrading macromolecules, hydrolysis zone assays were conducted on selective agar plates containing starch, pectin, cellulose, or protein as the sole substrate. Strains were spot-inoculated and incubated until visible colonies formed. For starch hydrolysis, the plates were stained with iodine solution, and the formation of a clear zone around the colony indicated amylase activity. Pectin-degrading activity was assessed by staining with 5 mg/mL Congo red solution for 15 min, followed by decolorization with 1 mol/L NaCl for another 15 min, with clear zones indicating positive results. For cellulose hydrolysis, plates were stained with 0.2% Congo red for 1 h and decolorized using 1 mol/L NaCl for 1 h, and the presence of transparent halos was used to indicate cellulase activity. Protein degradation was directly observed by the formation of a clear zone around the colonies on skim milk powder-containing agar plates. In all cases, the diameters of the hydrolysis zone (D) and colony (d) were measured, and the D/d ratio was calculated to assess the relative enzymatic potential.

## 3. Results and Discussion

### 3.1. Variations in Physicochemical Parameters and Major Chemical Constituents During Pu-erh Tea Pile Fermentation

Pile fermentation is a microbially driven process involving the oxidation, reduction, and catabolic transformation of tea constituents [24]. To investigate the environmental changes that influence microbial activity and metabolite production, we monitored key physicochemical parameters—including the temperature, pH, moisture content (%), and water activity (Aw)—throughout the 30-day fermentation of Little Brown Pu-erh tea. As shown in Figure 1a, the temperature of the tea pile rose sharply from 28.10 °C on day 0 to 49.63 °C on day 2, peaking at 51.50~62.90 °C from days 3 to 26. This thermogenic phase reflects vigorous microbial metabolism and biomass accumulation, a phenomenon that is common in compost-like fermentation systems. By day 30, the temperature had declined to 45.10 °C, likely due to the depletion of readily degradable substrates and the attenuation of microbial respiration in Little Brown Pu-erh tea [25,26]. These measurements were conducted at the middle layer of the pile, which served as the primary sampling zone. While this layer is often used to represent the overall fermentation dynamics, we acknowledge that temperature gradients may exist between the surface, core, and bottom regions of the pile, potentially affecting localized microbial activity and metabolite production. Meanwhile, the pH remained relatively stable, ranging from 6.0 to 7.0, with a minimum of 6.4 on day 2 and a maximum of 6.7 on day 14. Such near-neutral pH conditions are known to favor the activity of both bacterial and fungal communities, particularly *Bacillus* and *Aspergillus* ssp. The moisture content showed a biphasic trend (Figure 1b), increasing from 31.81% to 43.49% by day 5, potentially due to microbial respiration and water retention by biofilm matrices. This was followed by a gradual decline to 20.39% by day 30, consistent with water loss and reduced microbial activity. Similarly, A_W_ peaked at 0.99 on day 3 and declined to 0.93 by the end of fermentation, indicating the reduced availability of unbound water. Notably, despite the decline in the total moisture content after day 5, Aw remained relatively high due to the presence of unbound water retained within microbial biofilms and extracellular polymeric substances, which can maintain water availability and microbial activity in low-moisture conditions [14].

In parallel, major chemical components underwent substantial transformation. The concentrations of soluble polysaccharides, total flavonoids, theabrownin, protein, and lignin increased significantly over time, rising from 13.89, 8.79, 18.36, 275.63, and 316.67 mg/g to 22.31, 18.45, 60.90, 333.96, and 558.33 mg/g, respectively (Figure 1c,e,f). These increases suggest the microbial-induced degradation of cell wall polysaccharides and the biosynthesis of secondary metabolites, contributing to flavor- and health-related attributes. Conversely, tea polyphenols, free amino acids, and cellulose showed marked reductions, by 56.8% (107.22 mg/g), 35.5% (13.61 mg/g), and 33.9% (108.33 mg/g), respectively (Figure 1d–f), likely due to microbial utilization and enzymatic degradation. Together, these results demonstrate that pile fermentation not only modulates the physicochemical environment but also drives the bioconversion of key tea constituents. The observed trends lay the groundwork for subsequent microbial succession and metabolite production, emphasizing the importance of environmental monitoring and control in regulating the fermentation quality and consistency [27]. One limitation of this study is that the tea samples were collected exclusively from the middle layer of the fermentation pile. Although this layer is widely recognized as the most active zone in terms of microbial succession and enzymatic metabolism, it may not fully capture the vertical heterogeneity across the pile. Future studies are encouraged to incorporate stratified sampling across the top, middle, and bottom layers to better resolve the spatial microbial and metabolic differences under pile fermentation conditions.

### 3.2. Whole-System Microbial Community Profile During Pile Fermentation

#### 3.2.1. Alpha Diversity and Beta Diversity Analysis

Alpha diversity analysis revealed distinct temporal patterns in the prokaryotic and eukaryotic communities during pile fermentation. Specifically, a total of 39,995 clean reads for the 16S rRNA V3–V4 region and 44,159 clean reads for the fungal ITS1 region were generated, with average coverage of 99.58% and 95.99%, respectively. This depth was sufficient for downstream diversity and network analysis. In the prokaryotic microbiota (Figure 2a), richness and diversity metrics—including Chao1, Shannon, and Simpson indices—increased during the early stages (BPT_0D to BPT_22D), peaking on day 26 (Shannon = 0.91, *p* < 0.05), followed by a gradual decline toward day 30. This trend likely reflects the initial availability of abundant carbon sources that support microbial proliferation, followed by substrate depletion and ecological filtering favoring dominant taxa. Notably, the alpha diversity values of unfermented tea (BF_0D) were significantly higher than those of most fermented samples, indicating a richer but more heterogeneous microbial community at baseline. In contrast, the eukaryotic richness declined steadily throughout the fermentation process, although without statistically significant variations between stages, suggesting greater structural stability among fungal communities (Figure 2b).

Beta diversity analysis revealed the clear temporal structuring of the microbial communities during Pu-erh tea pile fermentation. The prokaryotic profiles from the middle to late stages (second to fifth fermentation cycles) exhibited strong clustering in the PCoA plots (Figure 2c,e), indicating increased homogeneity and ecological stabilization, with *Staphylococcus* (51.79%, 71.40%) and *Kocuria* (45.93%, 25.62%) as the dominant genera at BPT_28D and BPT_30D. The high structural similarity between these samples further confirmed the stability of the prokaryotic communities at the end of fermentation. Conversely, sample BPT_10D exhibited distinct separation from other clusters, likely due to the transient dominance of unclassified *Enterobacteriaceae* (35.31%). Eukaryotic communities displayed a different pattern, with stage-specific periodic clustering throughout fermentation. The PCoA analysis (Figure 2d,f) showed that the eukaryotic communities in the early stages (BF_0D-BF_10D) clustered together, as did those in the later stages (BF_12D-BF_30D), indicating temporal shifts in composition despite overall structural stability. Hierarchical clustering analysis further revealed intersample differences, as shown in the tree plot (Figure 2g,h, left) and the distribution of the top 10 genera (Figure 2g,h, right). At the eukaryotic level (Figure 2h), notable genera included *Aspergillus*, *Thermomyces*, *Rasamsonia*, and *Blastobotrys*, which were consistently abundant, although they varied in relative abundance across fermentation stages. In contrast, the prokaryotic composition (Figure 2g) showed more dynamic succession, with *Bacillus* dominating in the early stages and *Staphylococcus*, *Kocuria*, and *Brachybacterium* gradually becoming dominant later. Notably, such bacterial turnover contrasts with the relatively stable fungal structure. Interestingly, the opposite succession trends were observed in Kombucha fermentation, highlighting the distinct microbial ecology and environmental selection pressure unique to Pu-erh tea pile fermentation [28]. Further analysis of intergroup differences (Figure 2i,j) revealed that changes in prokaryotic composition were statistically significant, whereas eukaryotic distributions remained more consistent, supporting the interpretation that bacterial communities underwent substantial ecological restructuring, while fungal communities maintained compositional resilience throughout the fermentation process. In conclusion, these findings not only confirm well-established trends in microbial succession but also provide novel insights into the continuous evolution of the microbial communities throughout Pu-erh tea fermentation. By capturing detailed changes during the entire fermentation cycle, our work offers a more comprehensive view of the microbial dynamics and sets the stage for future studies focused on optimizing the fermentation process.

#### 3.2.2. Core Microbial Composition of Pu-erh Tea

Understanding the core microbial composition of Pu-erh tea is crucial in deciphering the microbial mechanisms underlying pile fermentation. Based on 15 representative time points, the integrated microbial profiles revealed diverse and dynamic microbial communities. First, we examined the relative abundance of microorganisms to characterize the profiles of Little Brown Pu-erh tea. In the microbial composition of the tea piling (Figure 3a), dominant prokaryotic genera included *Staphylococcus*, *Kocuria*, *Bacillus*, *Brachybacterium*, *Microbacterium*, *Brevibacterium*, *Pantoea*, and *Pluralibacter*. Notably, chloroplasts and mitochondria were also present in the composition due to the partial fragmentation of the tea leaves during metagenomic extraction. However, this did not alter the findings compared to the existing literature, which similarly identifies these genera as predominant. At the eukaryotic level (Figure 3b), significant genera included *Aspergillus*, *Thermomyces*, *Rasamsonia*, *Blastobotrys*, *Cyberlindnera*, *Rhizomucor*, *Debaryomyces*, *Candida*, *Aureobasidium*, and *Penicillium*. In our study, core taxa were defined as taxa with ≥1% relative abundance in the samples across the 15 fermentation time points. Among these, *Aspergillus* was particularly prominent across different stages due to its unique ability to produce amylase and cellulase, which are crucial in enhancing the taste and aroma of the tea, such as its sour flavor. Taxonomic hierarchy trees (Figure 3c,d) and phylogenetic composition plots (Figure 3e) provided further insights into the microbial distribution patterns at multiple taxonomic levels. These results collectively underscore the diverse and synergistic microbial consortia involved in Pu-erh tea fermentation, in which both prokaryotic and eukaryotic taxa play complementary roles in shaping the chemical and sensory characteristics of the final product. This comprehensive profiling of the core microbial genera forms a basis for targeted microbial regulation and functional validation in future studies on controlled fermentation.

#### 3.2.3. Species Difference and Marker and Association Network Analysis

Microorganisms perform a wide array of biological functions, secreting numerous enzymes during growth that catalyze various chemical reactions within the fermented tea pile [29]. To dissect species-level differences and intertaxa associations during Pu-erh tea pile fermentation, we performed integrated analyses combining Venn diagrams, LEfSe biomarker discovery, and microbial co-occurrence networks. The Venn analysis (Figure 4a,b) revealed five and six core species that were consistently present across all fermentation stages in the prokaryotic and eukaryotic communities, respectively. However, the number of unique taxa was markedly higher in prokaryotes (ranging from 58 to 2151) than in eukaryotes (1 to 118), suggesting greater ecological plasticity in bacterial communities.

Heatmaps displaying the top 20 genera (Figure 4c,d) highlighted temporal shifts in the community composition, where changes did not necessarily reflect total species turnover but rather the selective enrichment of specific taxa. The top genera were selected based on the clustering of the sample species composition using the Euclidean distance and the UPGMA algorithm. Species were clustered according to the Pearson correlation coefficient matrix, and the genera were arranged based on the resulting clusters. This approach ensured that the selected genera were representative of the microbial composition in the samples. LEfSe analysis identified statistically significant biomarkers differentiating samples across taxonomic levels (Figure 4e,f). Notable differences were observed in the BPT_12D sample at the prokaryotic level, particularly in taxa such as p_*Firmicutes*, c_*Bacilli*, and o_*Bacillales*. Conversely, robust differences were identified in the BPT_1D sample at the eukaryotic level, including taxa such as c_*Microbotryomycetes*, o_*Sporidiobolales*, f_*Sporidiobolaceae*, f_*Debaryomycetaceae*, g_*Debaryomyces*, and s_*Debaryomyces*_*prosopidis*. To provide more context regarding the ecological interactions among microbial taxa and their impacts on fermentation outcomes, we expanded the co-occurrence network analysis (Figure 4g,h). Our analysis revealed that eukaryotic taxa, particularly *Rasamsonia*, formed denser and predominantly positive associations compared to prokaryotes, suggesting a more cooperative role in shaping the microbial ecosystem. *Rasamsonia* exhibited the highest degree of connectivity, indicating its potential as a keystone species in fungal network stability and polysaccharide degradation. In contrast, prokaryotic communities showed both positive and negative correlations, reflecting dynamic competition and ecological filtering, which influenced the overall metabolic shifts during fermentation. For example, the Firmicutes and Bacilli genera exhibited different levels of dominance throughout fermentation, with shifts in abundance corresponding to changes in the available substrates. These findings provide insights into species succession, niche-specific differentiation, and the microbial consortia dynamics throughout fermentation.

#### 3.2.4. Functional Potential Prediction of Microbial Community in Different Periods

In microbial ecology, assessing not only the taxonomic composition but also the functional capacity of microbial communities is essential to understanding ecosystem dynamics. To infer the functional potential of the Pu-erh tea pile fermentation microbiota, we utilized PICRUSt2 to predict metagenomic profiles based on the 16S and ITS amplicon data. The predicted functional categories in prokaryotes encompassed biosynthesis, degradation/utilization/assimilation, detoxification, energy generation, glycan pathways, macromolecule modification, and metabolic clusters (Figure 5a). In contrast, eukaryotic communities lacked the detoxification and macromolecule modification categories (Figure 5b), indicating differential ecological roles between domains. Notably, prokaryotes exhibited higher functional richness and showed significant upregulation in pathways associated with energy metabolism and substrate degradation, which are crucial for biomass turnover and aroma compound release. To explore temporal shifts in microbial function, we compared four key fermentation stages (3D, 14D, 22D, and 30D), using unfermented tea (BF_0D) as a baseline. At 3D, few significant changes were observed, whereas, by 14D, prokaryotic pathways such as TEICHOICACID-PWY, LACTOSECAT-PWY, PWY-5304, and PWY-922 were markedly upregulated (Figure 5c), implicating active cell wall metabolism, carbohydrate degradation, and short-chain fatty acid biosynthesis—all of which contribute to tea flavor formation and microbial adaptation. In contrast, eukaryotic communities (Figure 5d) predominantly exhibited downregulation across fermentation stages, except for the upregulation of TYRFUMCAT-PWY, a pathway linked to tyrosine catabolism that may indirectly modulate volatile compound profiles. These functional predictions highlight domain-specific metabolic contributions during fermentation and offer mechanistic insights into the transformation of bioactive compounds and sensory traits in Pu-erh tea.

### 3.3. Metabolite Analysis of Pu-erh Tea Throughout Fermentation

To explore the temporal shifts in the metabolite composition and their potential associations with microbial dynamics, non-targeted metabolomics profiling was conducted on ten fermentation samples collected at 0, 1, 2, 3, 5, 6, 10, 12, 20, and 30 days. A total of 36,747 molecular features were detected, with 30,435 in positive ion mode and 6312 in negative ion mode. Based on HMDB and KEGG annotations, 273 and 74 metabolites were identified in positive and negative mode, respectively. These compounds included alkaloids, amino acids, fatty acids, peptides, shikimates, and phenylpropanoids (Figure 6a,b), representing a chemically diverse profile consistent with complex microbial biotransformation. Principal coordinate analysis (PCoA) revealed the clear clustering of early-stage samples (0D–6D) versus later-stage samples (10D–30D), indicating substantial shifts in metabolite profiles over time (Figure 6c,d). The high reproducibility of QC samples affirmed the reliability of the data. Based on this clustering pattern, we defined two groups for further analysis: early stage (ES) and late stage (LS). K-means clustering (based on VIP > 1 and *p* < 0.05) identified nine significant metabolite trends in positive mode (Figure 6e), with 2-acetyl-1-pyrroline and 2,5-dimethylpyrazine being key contributors to the popcorn- and roasted-like aromas accumulating prominently at 30D [30]. In negative mode (Figure 6f), seven trends were observed, including elevated levels of health-related unsaturated fatty acids such as arachidonic acid and linolenic acid in the LS group, both known for their anti-inflammatory properties [31]. Notably, the differential metabolite profiles appeared to stabilize toward the end of fermentation (20D–30D), suggesting the establishment of a relatively mature and functionally stable microbial ecosystem. Generally, these metabolites offer opportunities to improve the flavor profile and functional attributes of Pu-erh tea. For instance, controlling the fermentation conditions to enhance the production of these key metabolites could lead to more consistent flavor profiles and health benefits in industrial-scale tea production. These findings align with microbial succession patterns and provide metabolite-level evidence for quality development and potential functional benefits during Pu-erh tea fermentation [32].

### 3.4. Profile Analysis of Differential Metabolites

To establish reliable correlations between intergroup differences in metabolites and the experimental groups, irrelevant variables were further filtered out using OPLS-DA analysis. The results of the OPLS-DA model comparing the LS and ES groups are presented in Figure 7a,b. Here, the horizontal axis represents the predicted score of the first principal component, reflecting the intergroup differences between the LS and ES groups, while the vertical axis represents the orthogonal principal component score, indicating the intragroup differences among LS or ES samples. Notably, the types and quantities of metabolites differed between the LS and ES groups under both positive and negative ion mode. Furthermore, the homogeneity within the LS group was superior to that of the ES group.

The feasibility of the OPLS-DA model was further assessed through permutation testing, with the results depicted in Figure 7c,d. The horizontal axis in the figure indicates the permutation retention (with a retention value of 1), while R^2^Y and Q^2^ denote the values of the original OPLS-DA model. It is evident that the R^2^Y and Q^2^ values in the random model for both positive (A) and negative (B) ion mode decrease as permutation retention is reduced, suggesting that the original OPLS-DA model did not exhibit overfitting. Moreover, the model exhibited good interpretability (R^2^) and predictability (Q^2^), indicating that it accurately reflected the metabolite differences between ES and LS samples. Following this analysis, and in conjunction with the statistical results of unit and multivariate variables, differential metabolites between the LS and ES groups were identified with VIP > 1 and *p* < 0.05. As shown in Figure 7e, a total of 8133 material peaks exhibited significant differences between the LS and ES groups in positive ion mode, of which 4525 peaks were significantly upregulated and 3607 peaks were significantly downregulated compared to the ES group. In negative ion mode (Figure 7f), a total of 1985 material peaks showed significant differences between the LS and ES groups, including 1645 peaks that were significantly upregulated and 340 peaks that were significantly downregulated compared to the ES group.

### 3.5. Screening and Identification of Differential Metabolites

Subsequently, the HMDB and KEGG databases were utilized to annotate the differential substance peaks between the LS and ES groups, resulting in the identification of 37 and 19 metabolites in positive and negative ion mode, respectively. Based on this, the Euclidean distance matrix was calculated using the quantitative values of the differential metabolites between the two groups, followed by hierarchical clustering analysis. The differences in metabolite content between the groups were then visualized in a heatmap (Figure 8a,b). To analyze the most influential differential metabolites in the ES and LS groups, the corresponding ratios of the quantitative values of these metabolites were calculated and transformed using base 2 logarithms. The top 10 differential metabolites exhibiting up- and downregulation in each group are presented in Figure 8c,d. In positive ion mode (Figure 8c), the content of 4-hydroxytetradecanoic acid, xanthine, andrographolide, trolox, muscone, (−)-caryophyllene oxide, cyclosporine A, cinnamic acid, 2,5-dimethylpyrazine, and 12-oleanene-3,16,21,22,28-pentol was significantly upregulated during the late fermentation period (*p* < 0.05). Conversely, the content of 1(10), 4-cadinadiene, vinylpyrazine, 1,1,3-trimethylindene, dodecanoic acid, L-theanine, 4-hydroxybenzoic acid, obacunoic acid, 2-ethyl-3,5-dimethylpyrazine, 3,3′,4′,5,7-pentahydroxyflavone, and theobromine was significantly downregulated during this period (*p* < 0.001). In negative ion mode (Figure 8d), the content of linoleic acid, chlorogenic acid methyl ester, arachidonic acid, glycine anhydride, 3-hydroxy-3-methyl-2,4-nonanedione, thymol, soyasaponin I, 9,12-octadecadienoic acid, 5-megastigmene-3,9-diol, and cis-2-hexenyl hexanoate was significantly upregulated in the late fermentation period (*p* < 0.01). Notably, the changes in these metabolites are closely associated with tea quality, consistent with previous reports on tea flavor [33]. Specifically, cinnamic acid and linoleic acid contribute to aroma formation, with their significant upregulation enhancing the aroma of Pu-erh tea. Additionally, the significant increase in (-)-caryophyllene oxide during late fermentation is linked to the improved antioxidant properties of Pu-erh tea. Meanwhile, the notable downregulation of theobromine contributes to a reduction in the bitterness of Pu-erh tea.

### 3.6. Metabolic Pathway Analysis of Differential Metabolites

To further elucidate the enrichment of differential metabolites in specific metabolic pathways, their pathways were annotated using the KEGG PATHWAY database. As shown in Figure 9a, the metabolic pathways of caffeine metabolism and the biosynthesis of unsaturated fatty acids were the most enriched in the tertiary metabolic pathways under positive ion mode. In negative ion mode, differential metabolites were primarily involved in the biosynthesis of unsaturated fatty acids, linoleic acid metabolism, and cutin, suberine, and wax biosynthesis, as well as fatty acid biosynthesis (Figure 9b). The high abundance of metabolic pathways in both modes indicates that the metabolic activity of core bacteria was more vigorous during pile fermentation, highlighting the need to deconstruct this process. Notably, the high abundance of the caffeine metabolism pathway significantly contributes to a reduction in caffeine content in Pu-erh tea. This finding aligns with the roles of specific microbes, such as *Bacillus* and *Aspergillus*, which are known to contribute to the breakdown of complex molecules like caffeine and polyphenols, supporting the observed decrease in caffeine levels. These microbial aspects help to explain the reduction in caffeine levels observed during fermentation, which is important in improving the flavor profile of Pu-erh tea, as lower caffeine content is often associated with a smoother taste. Additionally, the biosynthesis of unsaturated fatty acids, which accompanies the accumulation of aromatic compounds, plays a crucial role in the aroma and taste profile of the tea [34]. Based on the differential metabolites identified in the KEGG metabolic pathway analysis, we further calculated the rich factor to analyze the enrichment degrees of the annotated differential metabolites relative to the total metabolites. As shown in Figure 9c, the rich factor for caffeine metabolism was the highest in positive ion mode, indicating a substantial proportion of differential metabolites involved in this pathway. Specifically, xanthine was significantly upregulated, while theobromine was significantly downregulated in comparison to unfermented samples. In negative ion mode (Figure 9d), the rich factor values for cutin, amber, and wax biosynthesis were the highest, suggesting that differential metabolites involved in these biosynthetic pathways exhibited the greatest proportions and degrees of enrichment. Specifically, oleic acid and long-chain fatty acids were significantly upregulated in the biosynthesis of cutin, suberine, and wax. The observed upregulation of these metabolites highlights the functional contribution of microbial communities, especially fungi like *Aspergillus*, which are linked to the production of aromatic compounds through the degradation of lipids. These metabolic shifts are significant for the complex flavor profile of Pu-erh tea, as they contribute to both the tea’s aroma and its potential health-promoting properties. Overall, these findings underscore the complexity of the Pu-erh tea fermentation process, highlighting the need for a comprehensive analysis of metabolite changes. The identification of key differential metabolites provides valuable insights into the roles of specific microbes in shaping the flavor, health benefits, and fermentation dynamics of Pu-erh tea. These metabolic pathways and microbial contributions will serve as important theoretical guidance in developing higher-quality Pu-erh tea.

### 3.7. Correlation Analysis Between Physicochemical Properties, Microbial Communities, and Metabolites of Pu-erh Tea

After analyzing the physicochemical properties, microbial communities, and differential metabolites of Pu-erh tea, mathematical models were constructed using RDA correlation analysis from the perspectives of prokaryotes and eukaryotes. As illustrated in Figure 10a, the prokaryotic communities during the fermentation of Little Brown Pu-erh tea were significantly influenced by water activity (63.5%, *p* = 0.002) and the temperature (23.5%, *p* = 0.01). Additionally, the pH exhibited a strong, positive correlation with the genus *Pluralibacter* in BPT_10D samples, while the moisture content was positively correlated with the Pseudomonas genus in BPT_2D samples. The prokaryotic flora also contributed significantly to the metabolism of free amino acids and tea polyphenols, with notable influences from water activity (67.1%, *p* = 0.002) and temperature (10.8%, *p* = 0.006). Furthermore, *Staphylococcus* showed a positive correlation with theabrownin (BPT_20D, BPT_22D, and BPT_26D), total flavonoids, soluble polysaccharides, lignin, and protein (BPT_14D) (Figure 10b), indicating that prokaryote taxa play a crucial role in the breakdown of macromolecules and the formation of important metabolites involved in flavor development. RDA analysis also revealed a significant correlation between tea polyphenols (54.2%) and the eukaryotic flora, highlighting the eukaryotic contribution to the metabolism of tea polyphenols during Pu-erh tea fermentation [35]. In the early stages, the total free amino acids, cellulose, water content, tea polyphenols, and water activity were strongly positively correlated with the dominant eukaryotic genera. Conversely, the pH, soluble polysaccharides, total flavonoids, theophycin, lignin, and protein were positively correlated with the dominant eukaryotic genera at later fermentation stages. Specifically, the relative abundance of *Aspergillus* was positively correlated with the content of free amino acids, tea polyphenols, and cellulose, while *Rasamsonia* and *Thermomyces* correlated positively with soluble polysaccharides, theophycin, lignin, protein, and total flavonoids. Additionally, the content of L-theanine, 4-hydroxybenzoic acid, 5-methyl-2-phenyl, and dodecanoic acid was positively correlated with the abundance of *Aureobasidium* and *Debaryomyces* (Figure 10c,d). The results from both the metabolic pathway analysis and RDA correlation models provide a more comprehensive understanding of how microbial communities shape the chemical composition of Pu-erh tea. These analyses clearly demonstrate that, while both prokaryotic and eukaryotic communities are involved in key metabolic transformations, the different roles that they play at various fermentation stages contribute to the overall complexity of the fermentation process. The functional activity of specific microbial taxa—such as the hydrolytic and oxidative enzymes of *Bacillus* and *Aspergillus*—is critical for the breakdown of macromolecules, the synthesis of essential metabolites, and the formation of the characteristic flavors and health-promoting properties of Pu-erh tea.

### 3.8. Functional Characterization of Core Microbial Strains from Pu-erh Tea Pile Fermentation

#### 3.8.1. Isolation and Taxonomic Identification of Representative Strains

To further investigate the functional roles of microorganisms involved in the pile fermentation of Pu-erh tea, we isolated and identified culturable microbial strains from different fermentation stages. Prokaryotic strains were cultured on LB agar medium, and colonies were purified by repeated streaking. Genomic DNA was extracted and subjected to PCR amplification of the 16S rRNA V3-V4 domain. The resulting sequences were compared with the NCBI database for taxonomic identification. A total of 17 prokaryotic strains were successfully identified, including *Bacillus haynesii*, *Staphylococcus gallinarum*, and *Bacillus licheniformis*, as summarized in Table 1. In parallel, eukaryotic fungi were enriched on PDA medium using serial dilution and streak isolation techniques. Single fungal colonies were selected for ITS amplification and sequencing. A total of 10 fungal strains were identified, such as *Aspergillus tubingensis*, *Aspergillus niger*, and *Hamigera insecticola*. Moreover, all microbial strains isolated and purified on agar plates in this study are visually documented in Appendix A. These isolates provide a valuable microbial resource for the study of fermentation mechanisms and serve as the foundation for subsequent functional validation experiments.

#### 3.8.2. Evaluation of Macromolecule Degradation by Representative Strains on Solid Media

To evaluate the metabolic potential of core strains during pile fermentation, six representative microorganisms were selected based on their growth performance, including four prokaryotes (Puer_*Bh*, Puer_*Bl*, Puer_*Bs*, and Puer_*Ba*) and two eukaryotes (Puer_*At* and Puer_*An*). In addition, these strains were selected for practical reasons, including their availability from the fermentation samples, ease of culture, and robust growth under fermentation conditions. The above strains were inoculated onto agar plates containing starch, pectin, cellulose, or protein as the sole substrate, and their degradation abilities were assessed based on the ratio of the hydrolysis zone diameter to the colony diameter, with a ratio >1 indicating significant enzymatic activity (Appendix A). Among prokaryotic strains, *B. haynesii* (Puer_*Bh*) exhibited the strongest cellulose-degrading ability (1.67), while *B. subtilis* (Puer_*Bs*) demonstrated the strongest protein degradation (2.14) and also led in starch (1.44) and pectin (1.64) breakdown, suggesting its broad-spectrum enzymatic activity and likely contribution to the production of soluble polysaccharides and free amino acids during fermentation. For the fungal strains, *A. tubingensis* (Puer_*At*) showed superior pectin degradation (3.85), whereas *A. niger* (Puer_*An*) was more effective in degrading cellulose (1.13), starch (1.26), and protein (1.38). These results highlight the distinct enzymatic profiles and functional capacities of the dominant strains, providing experimental evidence for their involvement in the biotransformation of macromolecules throughout the fermentation of Pu-erh tea (Figure 11). Unlike previous studies that focused solely on microbial profiling [11], our study employed a multi-omics approach, integrating microbial community dynamics with metabolite profiling. This provides a more holistic view of how microbial communities influence the biosynthesis of key flavor and bioactive compounds during fermentation. By combining microbial profiling with metabolomics and functional strain validation, we can link microbial activity directly to biochemical transformations, filling the gaps in understanding left by previous studies. In contrast tostudies that concentrated, on microbial succession [13], our study not only tracked microbial changes but also provides insights into their functional roles in shaping the flavor and health profile of Pu-erh tea. This integrated analysis allows for a deeper understanding of the fermentation dynamics than has been previously possible. However, the functional study of the dominant strains in Pu-erh tea fermentation is a broader task, and further research will be needed to explore the full spectrum of microbes involved in this complex process. Collectively, the enzymatic activity of these isolates supports the observed shifts in key metabolite levels and lays the groundwork for future co-culture studies and fermentation trials aimed at optimizing tea quality through targeted microbial interventions.

## 4. Conclusions

As is well known, the microbial composition, collective functional genes, and flavor compounds change significantly during the fermentation process of Pu-erh tea [29,30]. This study provides an integrated understanding of microbial succession, metabolite dynamics, and functional microbial roles during the 30-day pile fermentation of Pu-erh tea. Through high-throughput sequencing and untargeted metabolomics, we revealed that microbial community transitions—particularly the shift from Bacillus to Staphylococcus and Kocuria in prokaryotes and the stable dominance of Aspergillus among eukaryotes—coincide with changes in physicochemical parameters and the production of flavor- and function-related metabolites. Notably, we identified several important differential metabolites, which are enriched in the biosynthesis of fatty acids and the metabolism of caffeine, playing key roles in both the flavor and health-promoting properties of Pu-erh tea. The correlation analysis demonstrated strong interactions among environmental factors, the microbial composition, and metabolic outcomes. Crucially, the functional validation of six representative strains from Bacillus and Aspergillus confirmed their enzymatic activity in degrading starch, pectin, cellulose, and protein, providing direct evidence for microbial contributions to macromolecular biotransformation during fermentation. Unlike previous descriptive studies, our work bridges predicted microbiota function with experimentally verified microbial activity, offering novel mechanistic insights into microbial–chemical interactions in dark tea fermentation. Furthermore, future research could focus on further exploring the functional characteristics of these strains, particularly under different fermentation conditions. Additional strains could be introduced and applied in the fermentation of other food products to further explore the enzymatic activity and metabolic pathways of these strains. This will help to validate their unique characteristics compared to known strains from other fermentation systems and provide a deeper understanding of their functional roles in Pu-erh tea fermentation. In conclusion, these findings support the development of targeted microbial interventions to improve the fermentation consistency, flavor formation, and product standardization in Pu-erh tea manufacturing.

## Figures and Tables

**Figure 1 microorganisms-13-01857-f001:**
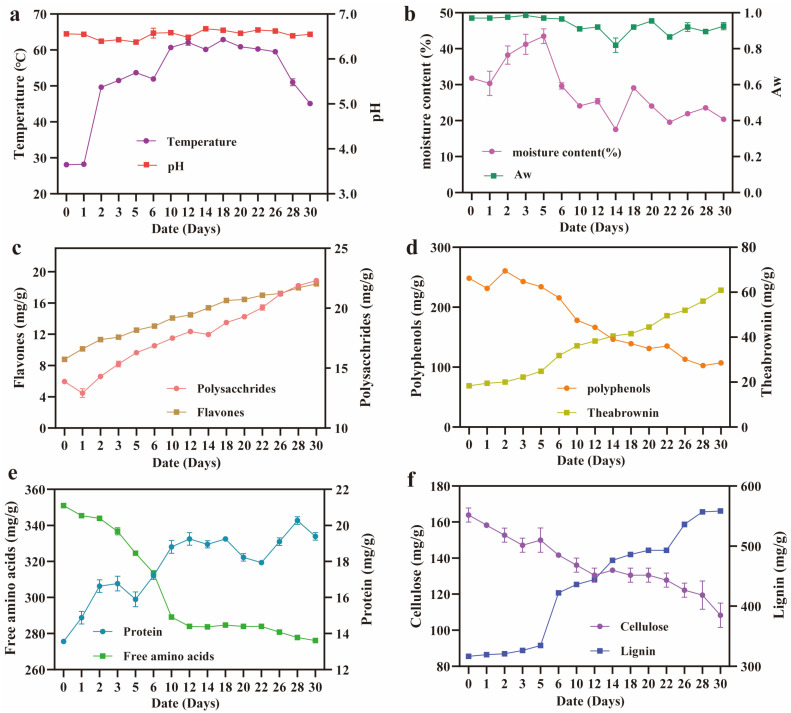
**Determination of physicochemical parameters and main chemical components of Pu-erh tea during pile fermentation.** (**a**) pH and temperature. (**b**) Moisture content (%) and water activity (Aw). (**c**) Polysaccharide and flavone content (mg/g). (**d**) Polyphenol and theabrownin content (mg/g). (**e**) Protein and free amino acid content (mg/g). (**f**) Cellulose and lignin content (mg/g). All experiments were repeated in triplicate.

**Figure 2 microorganisms-13-01857-f002:**
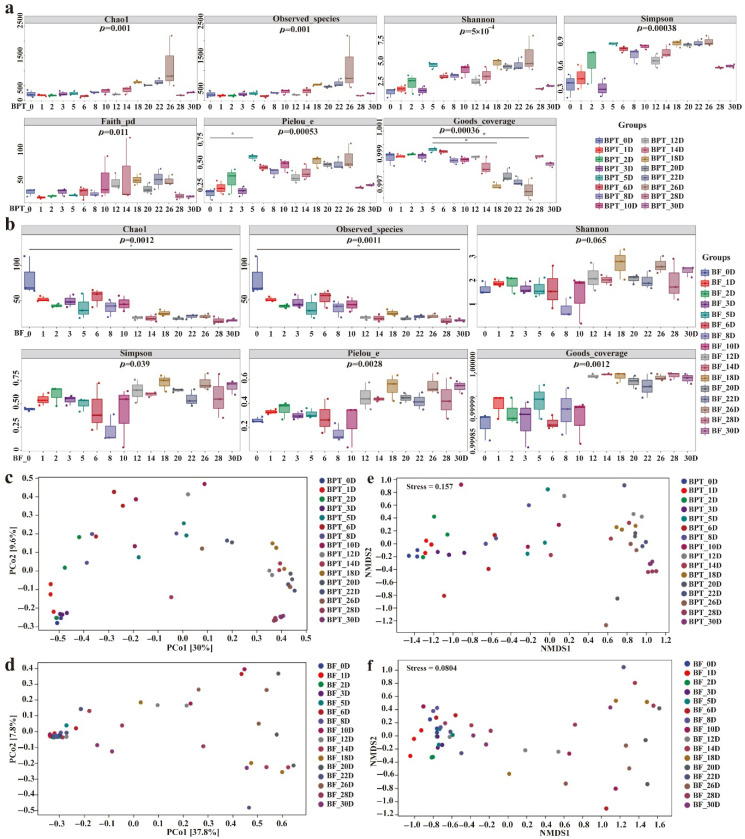
**Alpha and beta diversity analysis of Pu-erh tea throughout fermentation.** Alpha diversity indices of prokaryotic (**a**) and eukaryotic (**b**) microbial communities, with main parameters encompassing Chao1, observed species, Shannon, Simpson, Faith’s PD, Pielou’s evenness, and Good’s coverage. Distance matrix and PcoA analysis of prokaryotic (**c**) and eukaryotic (**d**) microbial communities. Non-metric multidimensional scaling analysis (NMDS) of prokaryotic (**e**) and eukaryotic (**f**) microbial communities. Hierarchical clustering analysis of prokaryotic (**g**) and eukaryotic (**h**) microbial communities. Analysis of differences between periods of prokaryotic (**i**) and eukaryotic (**j**) microbial communities (the sample at 0D was the control).

**Figure 3 microorganisms-13-01857-f003:**
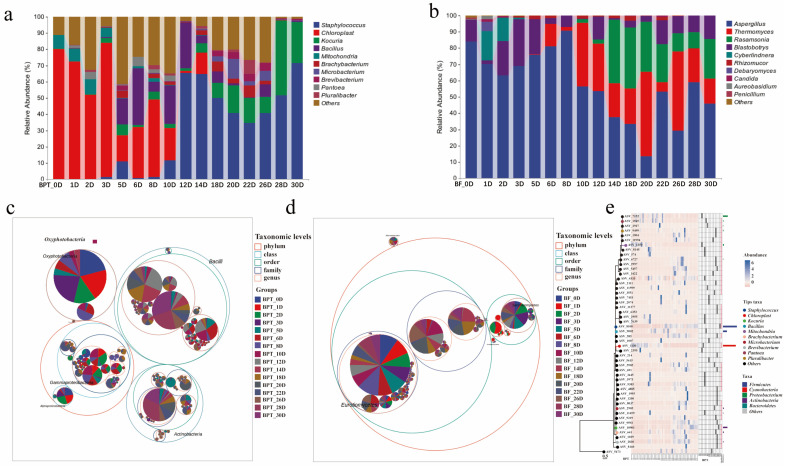
**Relative abundance of microbial communities of Pu-erh tea during different periods.** Bar charts of prokaryotic (**a**) and eukaryotic (**b**) microbial communities at the genus level. Classification level trees of prokaryotic (**c**) and eukaryotic (**d**) microorganisms. (**e**) Phylogenetic tree plot of prokaryotic communities.

**Figure 4 microorganisms-13-01857-f004:**
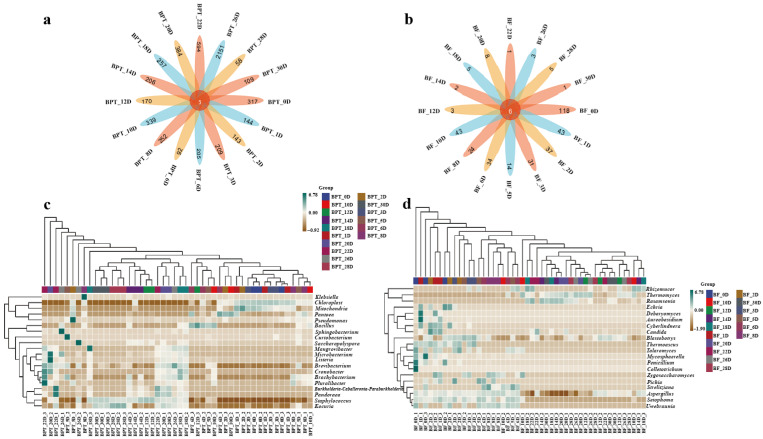
**Analysis of species differences and marker and association network analysis during pile fermentation.** Venn diagrams of prokaryotic (**a**) and eukaryotic (**b**) microbial communities. Species composition heatmaps of prokaryotic (**c**) and eukaryotic (**d**) microbial communities. LEfSe analyses of prokaryotic (**e**) and eukaryotic (**f**) microbial communities. LDA threshold was set at 2. Association networks of prokaryotic (**g**) and eukaryotic (**h**) microbial communities. Sample names, such as BF_28D and BPT_28D, represent prokaryotic (BF) and eukaryotic (BPT) microbial communities, respectively, with numbers following “D” indicating time period of Pu-erh tea fermentation (28D refers to samples collected on 28th day of fermentation).

**Figure 5 microorganisms-13-01857-f005:**
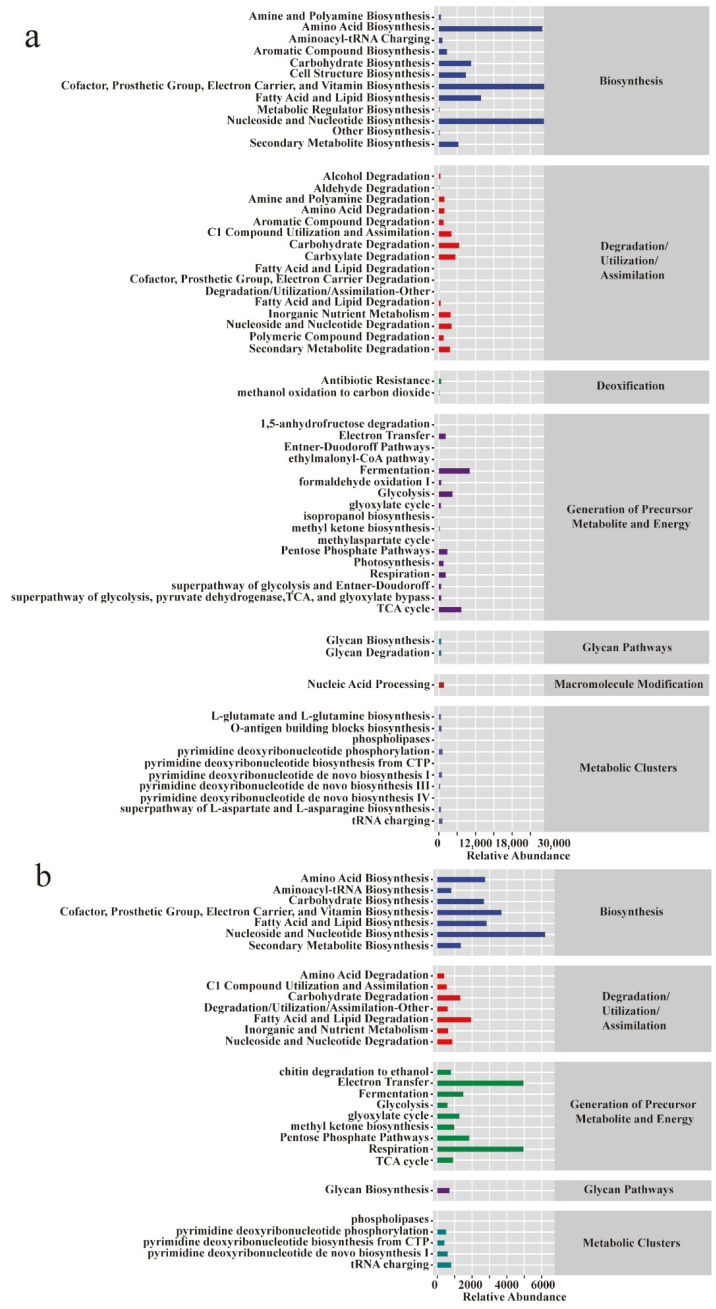
**Functional potential prediction of the microbiome of Pu-erh tea during pile fermentation.** Metabolic pathway statistics involved in prokaryotic (**a**) and eukaryotic (**b**) microbial communities. Differential analysis of metabolic pathways at various stages in prokaryotic (**c**) and eukaryotic (**d**) microbial communities. The sample at 0D was set as a control; other groups were used to investigate the change pathways (upregulated and downregulated).

**Figure 6 microorganisms-13-01857-f006:**
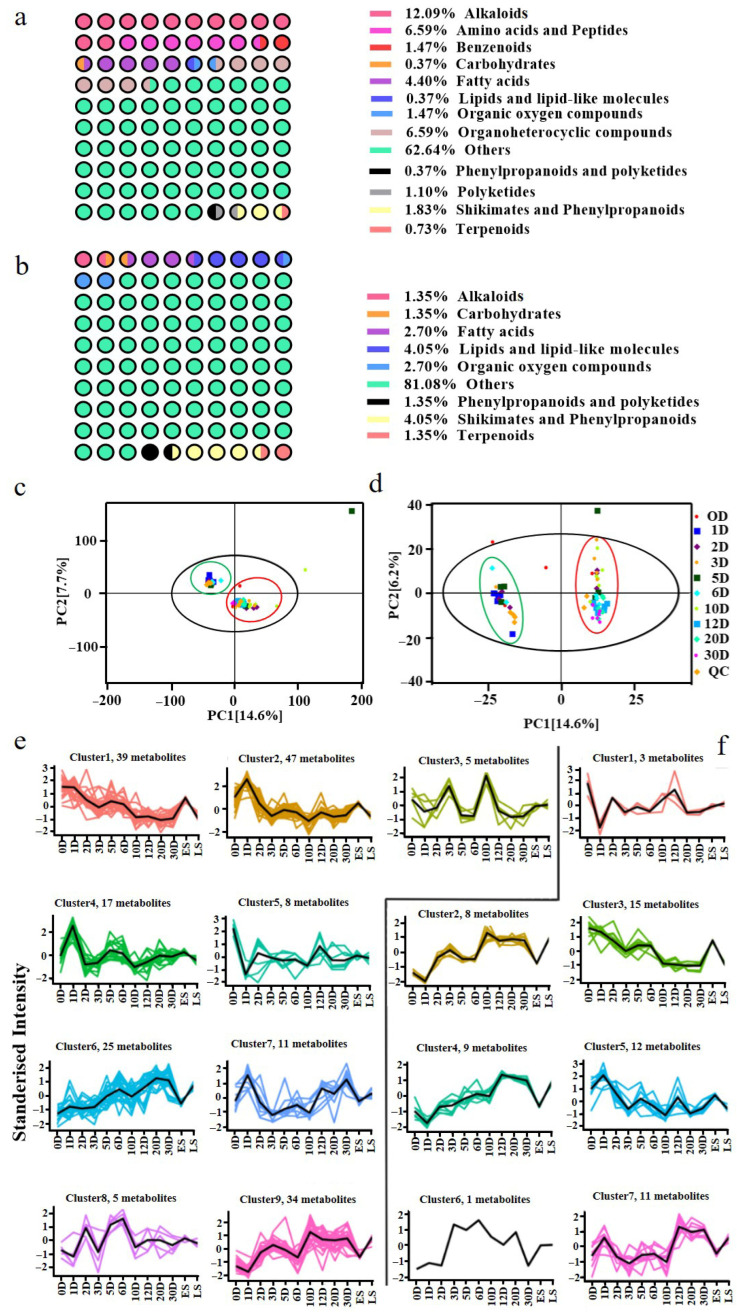
**Metabolite analysis of non-targeted metabolomics.** The percentages of different types of metabolites in positive (**a**) and negative (**b**) ion mode. PCoA plots of total sample metabolites in positive (**c**) and negative (**d**) ion mode. K-mean clustering analysis of positive (**e**) and negative (**f**) ion mode.

**Figure 7 microorganisms-13-01857-f007:**
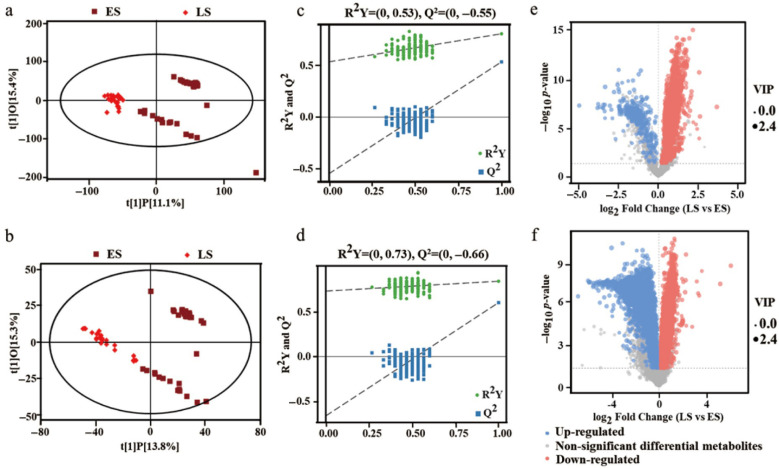
**Profile analysis of differential metabolites.** The OPLS-DA plots of metabolites in LS to ES in positive (**a**) and negative (**b**) ion mode. The OPLS-DA model substitution test for metabolites in LS to ES in positive (**c**) and negative (**d**) ion mode. Volcano plots of differential metabolites in ES and LS in positive (**e**) and negative (**f**) ion mode. Group ES (early stage) represents samples at 0~6D; Group LS (later stage) represents samples at 10~30D.

**Figure 8 microorganisms-13-01857-f008:**
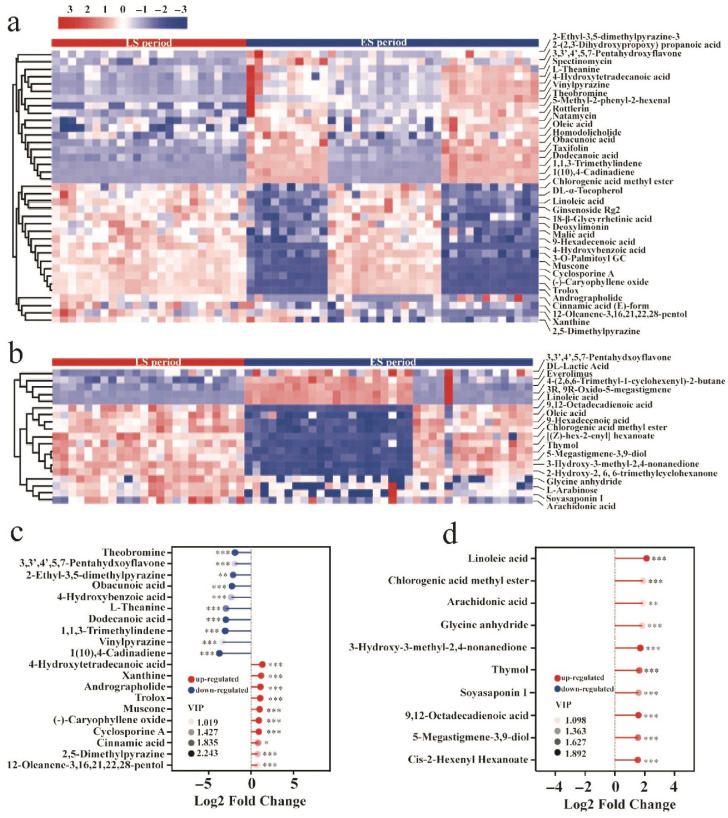
**Screening and identification of differential metabolites.** The clustering results for differential metabolites in ES/LS in positive (**a**) and negative (**b**) ion mode. The tinder plots of the differential metabolites in LS to ES in positive (**c**) and negative (**d**) ion mode. *: *p* < 0.05; **: *p* < 0.01; ***: *p* < 0.001.

**Figure 9 microorganisms-13-01857-f009:**
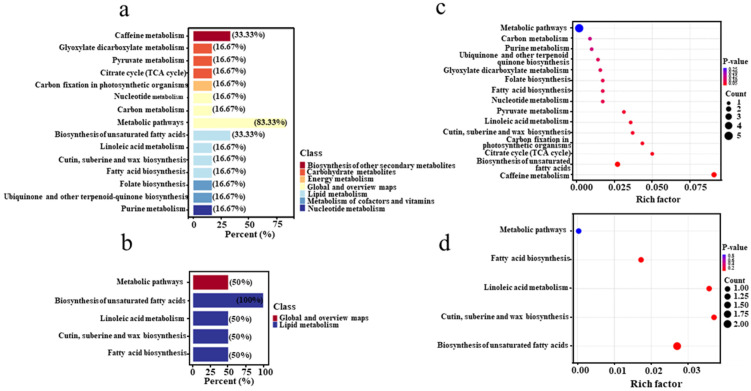
**Enrichment of KEGG metabolic pathways**. Metabolic pathway classification of differential metabolites in ES/LS in positive (**a**) and negative (**b**) ion mode. KEGG enrichment maps of differential metabolites in ES and LS in positive (**c**) and negative (**d**) ion mode.

**Figure 10 microorganisms-13-01857-f010:**
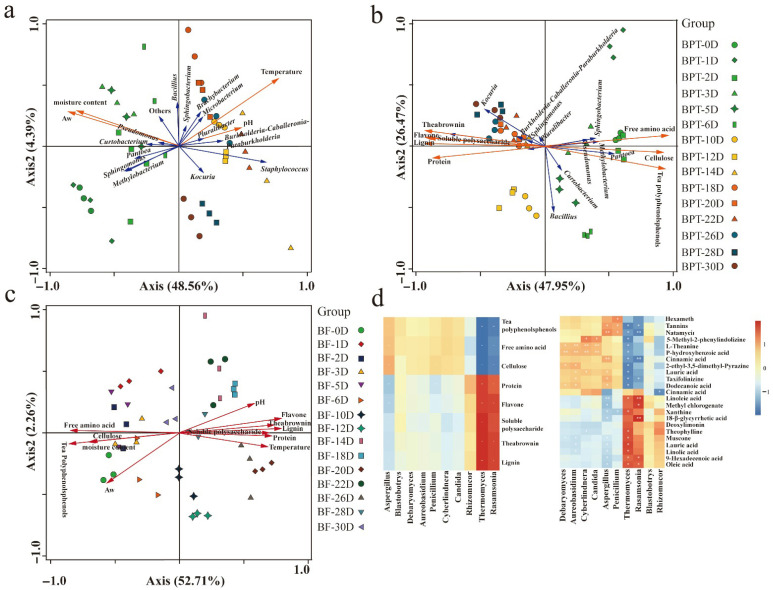
**Correlation analysis among microbial composition and biochemical parameters and metabolites** The redundancy analysis of the dominant prokaryotic genera in BPT with biochemical parameters (**a**) and the main chemical components (**b**). The redundancy analysis of the dominant eukaryotic genera and biochemical indicators (**c**). Correlations between microbiome and main chemical components (**d**). * indicates *p* < 0.05; ** indicates *p* < 0.01.

**Figure 11 microorganisms-13-01857-f011:**
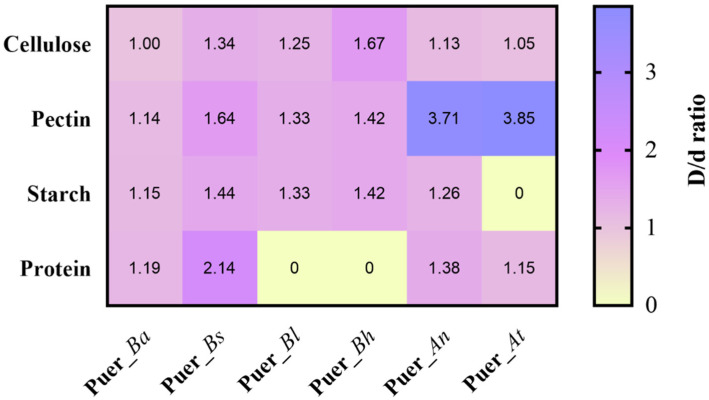
**Biodegradation screening of macromolecules by representative bacterial and fungal strains.** Agar plate hydrolysis assays were conducted using carboxyl methyl cellulose, skim milk powder, starch, and pectin as substrates. D: hydrolysis ring diameter; d: colony diameter; D/d ratio > 1 indicates enzymatic activity.

**Table 1 microorganisms-13-01857-t001:** Taxonomic identification and strain abbreviations of microbial isolates obtained from Pu-erh tea pile fermentation.

Species	Identity (%)	Strain Abbreviation
Prokaryotes		
*Bacillus haynesii*	99.66%	Puer_*Bh*
*Bacillus licheniformis*	99.93%	Puer_*Bl*
*Bacillus subtilis*	99.59%	Puer_*Bs*
*Bacillus amyloliquefaciens*	99.93%	Puer_*Ba*
*Staphylococcus gallinarum*	99.98%	Puer_*Sg*
*Aeromonas caviae*	99.86%	Puer_*Ac*
*Priestia aryabhattai*	99.80%	Puer_*Pa*
*Curtobacterium citreum*	99.51%	Puer_*Cc*
*Priestia filamentosa*	99.76%	Puer_*Pf*
*Rothia halotolerans*	99.52%	Puer_*Rh*
*Lysinibacillus macroides*	99.45%	Puer_*Lm*
*Staphylococcus lloydii*	99.79%	Puer_*Sl*
*Pluralibacter gergoviae*	99.51%	Puer_*Pge*
*Enterococcus faecium*	99.73%	Puer_*Ef*
*Pseudomonas guariconensis*	98.49%	Puer_*Pgu*
*Klebsiella pneumoniae*	99.31%	Puer_*Kp*
*Mammaliicoccus sciuri*	99.86%	Puer_*Ms*
Eukaryotes		
*Aspergillus tubingensis*	99.82%	Puer_*At*
*Aspergillus niger*	99.82%	Puer_*An*
*Rhizomucor pusillus*	99.66%	Puer_*Rp*
*Cyberlindnera rhodanensis*	99.13%	Puer_*Cr*
*Blastobotrys adeninivorans*	99.47%	Puer_*Ba*
*Trichosporon asahii*	98.00%	Puer_*Ta*
*Aspergillus costaricensis*	99.00%	Puer_*Ac*
*Hamigera fusca*	99.00%	Puer_*Hf*
*Hamigera insecticola*	100.00%	Puer_*Hi*
*Lichtheimia corymbifera*	98.00%	Puer_*Lc*

## Data Availability

The original contributions presented in this study are included in the article/Appendix A. Further inquiries can be directed to the corresponding authors.

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
