# Peer review of "Integrated Microbiome–Metabolome Analysis and Functional Strain Validation Reveal Key Biochemical Transformations During Pu-erh Tea Pile Fermentation"

_microorganisms, 2025, doi:10.3390/microorganisms13081857_

Round 1

Reviewer 1 Report

Comments and Suggestions for Authors

The contribution of Hu et al reports an interesting and polyphasic work on the microbial, metabolic, and functional profiling of the Pu-erh tea fermentation pile. However, several major and minor concerns emerged during the review of this manuscript version that must be addressed before it can be accepted for publication.  Particularly, as other reports have been published previously, precisely on the microbial and metabolic characterization of Pu-erh tea fermentation, the large amount of work performed for this contribution loses its relevance if it is not highlighted properly with respect to previous publications.  

Lines 116-117: 200g tea leaves (Fig. S2) were selected from each of 5 points A, B, C, D and E in the middle layer of the tea stack,…” Did you assume that the fermentation conditions are similar in the middle section of the pile to those in the bottom or on the surface? Does sampling location affect the type of microorganisms isolated and those detected by amplicon sequencing? 
Lines 273-275: Is the temperature reported homogeneous in all the zones of the entire pile? If you do not accurately determine the temperature at the bottom of the pile, the reported values correspond to the sampled middle zone. How can this gradient temperature affect the microbial and chemical changes in the system during fermentation?
In Figure 1B, how do you explain that if Aw remains relatively constant during fermentation, the moisture content decreases from hour 5?
Section 3.2.1: Precise the amount of high-throughput sequencing high-quality data generated for bacterial 16S rRNA V3–V4 region and or fungal ITS1 region. 
Figure 2. Alpha and Beta diversity analysis of the Pu-erh tea across fermentation is extensive. Consider including part of this figure as supplementary material or splitting it into two figures. 
Lines 363-373, I agree that the identified OTUs could be considered the microbial core of Pu-erh tea fermentation; however, my primary concern is the similarity in this core if you sampled another region of the pile, e.g., the bottom. Precise the threshold value applied to define core relative abundances.
This concern also applies to the metabolite profile analysis reported. Could this profile be the same among all the fermentation pile sections? (Sections 3.3 and 3.4).
Line 397: Defines the criteria to select the top genera. 
In Figures 5 and 8 (panels a and b), it isn't easy to read the information on the X axis. Consider modifying the information included in these figures to improve proper analysis. 
Compare and discuss the information in the Introduction and Results and Discussion sections and the relevance of your contribution with previous reports on the analysis of the Pu-erh tea: https://doi.org/10.1016/j.lwt.2022.113128, 10.1371/journal.pone.0157847, https://doi.org/10.1038/srep10117.

Minor concerns

Line 51: Include additional references supporting the beneficial properties of Pu-erh tea, as the authors describe several relevant health benefits. 
Line 102: “Ltd., the primers used…”, change to: “Ltd. The primers used…”  Indicate here that the sequence of primers used is presented in Supplementary Table 2. 
Line 224, 239: “First,1.0 g”, change to First, 1.0 g  
Line 236: “Cellulose”, change to cellulose
Line 282: Change “spp” to “spp”. It is not a taxonomically valid term. 
Line 306: flavones
Lines 328, 583: prokaryotic flora”. Flora refers to the plants of a particular region, habitat, or geological period.  Today, it does not apply to microbial diversity analysis.
Line 417: species.
Section 3.6: Metabolic…
Line 663: Delete of Pu-erh tea. Avoid redundancies (line 661).

Reviewer 2 Report

Comments and Suggestions for Authors

This study tracked physicochemical changes, microbial succession, and metabolite profiles across 15 fermentation stages. Dominant microbes included Bacillus and Aspergillus, with prokaryotes showing greater diversity. Metabolomics revealed 347 compounds, including aroma and health-related metabolites enriched in later stages. Statistical analyses confirmed microbial-metabolite-environment correlations. Functional assays of isolated strains validated enzymatic roles in macromolecule degradation. This multi-omics approach offers mechanistic insights and supports microbial optimization in Pu-erh tea processing.

This is a well-performed study. But several points should be addressed.

Fig4 Sample names such as BF_28D should be described in the legend.
Fig2 and 5 sample names are invisible

Discussion I see many articles on pu-erh tea and microbes when searched with google scholar. The results obtained in this study should be compared with these and discussed.

P19 l669 Conclusion specific compounds should be described.

Reproducibility of the fermentation processes should be discussed.

Reviewer 3 Report

Comments and Suggestions for Authors

The study presents a comprehensive and well-integrated multi-omics approach, combining microbiome analysis, metabolomics, and functional strain validation to elucidate the biochemical transformations during Pu-erh tea pile fermentation. This holistic methodology provides valuable mechanistic insights into how microbial communities and metabolites evolve over time, linking specific microbial taxa with biochemical changes relevant to flavor and quality. The inclusion of both prokaryotic and eukaryotic communities, along with functional validation through enzymatic assays, strengthens the study’s conclusions and enhances its significance for both academic researchers and industry professionals interested in fermented foods and microbial ecology.

The manuscript’s main weaknesses lie in its presentation and clarity. The writing requires careful language editing to improve readability, reduce redundancy, and ensure consistency in terminology. Some sections of the methods and results are overly detailed and could benefit from more concise and structured presentation. Figures and tables, while comprehensive, sometimes lack clarity in labeling and explanation, making it harder for readers to quickly grasp key findings. Additionally, while the study design is robust, the discussion could better highlight the broader implications of the findings for industrial applications and future research directions.

Specific recommendations:

  • Provide a clearer explanation of how the identified microbial taxa specifically contribute to the observed metabolic shifts; the current link between microbial function and metabolite changes is not always well substantiated.

  • Strengthen the interpretation of the metabolomics data by discussing possible metabolic pathways in more depth, particularly how these pathways relate to tea flavor, health benefits, and fermentation dynamics.

  • Clarify the novelty of findings regarding microbial succession; differentiate which observations confirm previous knowledge and which provide new insights into Pu-erh tea fermentation.

  • Expand on the significance of isolating specific strains (e.g., Bacillus, Aspergillus) — explain whether these strains show any unique characteristics compared to known strains from similar environments or fermentations.

  • Address the functional relevance of the identified enzymes more explicitly — discuss whether the observed hydrolytic activities are sufficient to explain the changes in tea chemistry or if other microbial actions are likely involved.

  • Justify the selection of six representative strains for validation; explain whether these strains are dominant, functionally important, or selected for practical reasons.

  • Discuss more critically whether the findings from laboratory-scale fermentations reflect industrial-scale processes, including environmental variability and microbial competition.

  • Provide more context on the ecological interactions among microbial taxa; co-occurrence network analysis is presented but not deeply interpreted in terms of microbial ecology or fermentation outcomes.

  • Address the limitations of predictive functional profiling (e.g., PICRUSt2) and validate these predictions where possible with metabolomics or enzymatic data.

  • Include a clearer rationale for why certain metabolites (e.g., aroma compounds, fatty acids) were prioritized in the analysis; discuss whether these findings offer actionable targets for process optimization.

Minor comments:

Page 1: "degradation1. Introduction" should have a space/new line

Line 305: Water activity (Aw) -->Should the w be subscript?

Line 306 (2x): Polysacchrides-->Polysaccharides

Fig. 6: alkaloid -->alkaloids

Comments on the Quality of English Language

The English is understandable, a native speaker should go over it for best formulations.

Round 2

Reviewer 1 Report

Comments and Suggestions for Authors

All the concerns and suggestions made during the first review process were addressed satisfactorily. 

Reviewer 3 Report

Comments and Suggestions for Authors

The reviewer comments were fully addressed.

I have no further remarks. This has become a good paper!